# *Tsc1* Loss in VIP-Lineage Cortical Interneurons Results in More VIP+ Interneurons and Enhanced Excitability

**DOI:** 10.3390/cells13010052

**Published:** 2023-12-26

**Authors:** Jia Sheng Hu, Ruchi Malik, Vikaas S. Sohal, John L. Rubenstein, Daniel Vogt

**Affiliations:** 1Department of Psychiatry, University of California San Francisco, San Francisco, CA 94158, USA; 2Weill Institute for Neurosciences, University of California San Francisco, San Francisco, CA 94158, USA; 3Center for Integrative Neuroscience, University of California San Francisco, 1550 4th St., San Francisco, CA 94158, USA; 4Sloan-Swartz Center for Theoretical Neurobiology, University of California San Francisco, 1550 4th St., San Francisco, CA 94158, USA; 5Department of Pediatrics and Human Development, Michigan State University, Grand Rapids, MI 49503, USA; 6Neuroscience Program, Michigan State University, East Lansing, MI 48824, USA

**Keywords:** *Tsc1*, MTOR, cortical interneuron, CGE, VIP, apoptosis, excitability

## Abstract

The mammalian target of rapamycin (mTOR) signaling pathway is a powerful regulator of cell proliferation, growth, synapse maintenance and cell fate. While intensely studied for its role in cancer, the role of mTOR signaling is just beginning to be uncovered in specific cell types that are implicated in neurodevelopmental disorders. Previously, loss of the *Tsc1* gene, which results in hyperactive mTOR, was shown to affect the function and molecular properties of GABAergic cortical interneurons (CINs) derived from the medial ganglionic eminence. To assess if other important classes of CINs could be impacted by mTOR dysfunction, we deleted *Tsc1* in a caudal ganglionic eminence-derived interneuron group, the vasoactive intestinal peptide (VIP)+ subtype, whose activity disinhibits local circuits. *Tsc1* mutant VIP+ CINs reduced their pattern of apoptosis from postnatal days 15–20, resulting in increased VIP+ CINs. The mutant CINs exhibited synaptic and electrophysiological properties that could contribute to the high rate of seizure activity in humans that harbor *Tsc1* mutations.

## 1. Introduction

Tuberous Sclerosis Complex (TSC) is a syndrome caused by mutations in either the *Tsc1* or *Tsc2* genes [1,2]. While TSC impacts multiple organ systems, those diagnosed also exhibit TSC-Associated Neuropsychiatric Disorders (TANDs) [3], of which associated cellular and molecular changes in the brain are incompletely understood. Notably, up to 90% and 40% of those diagnosed with TSC will have epilepsy and autism spectrum disorder (ASD), respectively [4,5,6,7]. Thus, understanding how *Tsc1* and *Tsc2* dysfunction impacts brain development and function is crucial.

*Tsc1* and *Tsc2* encode proteins that form an obligate complex that inhibits the activity of the mammalian target of rapamycin (MTOR) via its action on Rheb GTPase [8]. MTOR is often activated in response to growth factors and regulates multiple cellular events, including growth, proliferation and survival [9]. Importantly, *Tsc1*/*2*, as well as other signaling proteins in this pathway, are necessary to constrain MTORC1 activity when not required. Moreover, many genes that regulate MTORC1 activity, including *Tsc1*/*2*, are considered high-risk ASD candidate genes [10,11]. Thus, understanding their normal function in brain development and how their dysfunction impacts different brain cell types could provide new inroads into ASD biology and how other cognitive changes may arise. Specifically, uncovering how TANDs arise will require an understanding of *Tsc1*/*2*’s unique roles in the multitude of brain cell types.

The effect of *Tsc1*/*2* mutations on GABAergic cortical interneurons (CINs) is understudied. Due to their diverse molecular, morphological and electrophysiological properties [12,13,14], mutations in *Tsc1*/*2* could induce a variety of effects on cognition due to how different CINs are impacted. The majority of CINs are generated in the medial or caudal ganglionic eminences (MGE and CGE) and migrate into their cortical destinations in multiple waves from mid-gestation to early postnatal ages in rodents [15]. CINs can be divided into four broad groups based on the expression of molecular markers. MGE-lineage CINs primarily express either Somatostatin (SST) or Parvalbumin (PV), while CGE lineages express either vasoactive intestinal peptide (VIP) or neuron-derived neurotropic factor (NDNF) [16,17]. In addition, each broad group has distinct morphological and electrophysiological properties, with SST+ CINs targeting the dendrites of excitatory neurons and PV+ CINs targeting either the cell body or axon initial segments and exhibiting fast action potential kinetics [18]. CGE-lineage NDNF CINs, i.e., neurogliaform cells, are enriched in upper layers of the neocortex and can bulk release GABA, while VIP+ CINs mostly reside in upper cortical layers and have a bipolar morphology; they also can uniquely inhibit other CINs, providing a disinhibitory circuit in the cortex [19,20]. While our previous work found changes in MGE-lineage CIN properties after manipulating the mouse *Tsc1* gene and human variants [21,22], how CGE lineages are impacted by the same genetic perturbation has not been examined.

Thus, we used a Cre-driver line to delete mouse *Tsc1* in VIP CINs and assess their development. We unexpectedly found that WT CGE CINs have a delayed apoptotic cell loss, which differs from the timing of programmed apoptosis of MGE CINs [23]. In the *Tsc1* mutant, we found that there was reduced apoptosis of VIP+ CINs, resulting in more VIP+ CINs. In addition, loss of *Tsc1* in VIP+ CINs results in increased excitatory drive onto the VIP+ CINs. Overall, the increase in VIP CIN number and their increased excitability is likely to decrease cortical inhibition, thereby causing major imbalances in excitation to inhibition, a hallmark of ASD and epilepsy.

## 2. Materials and Methods

### 2.1. Animals

The following mouse strains have been previously reported: tdTomato lox/+ (Ai14) Cre-reporter [24], *Tsc1* lox/+ [25] and VIP-IRES-Cre [26]. All strains were on a mixed C57BL/6 and CD1 background. Animals were housed in a vivarium with a 12 h light/12 h dark cycle. Postnatal animals used for experiments were kept with their littermates. Males and females were both used in all experiments; no sex differences were observed. All animal care and procedures were performed according to the University of California at San Francisco Laboratory Animal Research Center guidelines. Mice were anesthetized with intraperitoneal avertin (0.015 mL/g of a 2.5% solution) followed by pneumothorax and perfused transcardially with PBS and then with 4% PFA, followed by brain isolation, 1 h fixation for synapse immunohistochemistry or overnight fixation for all other immunohistochemistry or in situ hybridization, and 2 days of cryoprotection with 30% sucrose and microtome sectioning (coronal, 40 μm).

### 2.2. Antibodies

Antibodies used were rabbit cC3 (1:250, Catalog #9661S, Cell Signaling Technologies, Danvers, MA, USA), rabbit DsRed (1:500, Catalog #632496, Takara, Mountain View, CA, USA), mouse nAChR α4 Subunit Antibody, clone 369 (1:50, Catalog #MABN1833, MilliporeSigma, St. Louis, MI, USA), rabbit phosphoS6^SER240/244^ (1:400, Catalog #5364, Cell Signaling Technologies, Danvers, MA, USA), mouse PSD95 (1:200, Catalog #75-028, NeuroMab, Davis, CA, USA), goat Vesicular Acetylcholine Transporter (VAChT) antibody (1:500, Catalog #ABN100, MilliporeSigma, St. Louis, MI, USA), rabbit Vglut1 (1:500, Catalog #135303, Synaptic Systems, Goettingen, Germany), rabbit VIP (1:250, Catalog #20077, Immunostar, Hudson, NY, USA) and Alexa-conjugated secondary antibodies (1:300, Thermo Fisher, Eugene, OR, USA). Sections were cover-slipped with Vectashield (Vector Labs, Burlingame, CA, USA).

### 2.3. Dual RNAscope (Fluorescent In Situ Hybridization) and IHC Labeling 

In situ hybridization (ISH) was performed on 40 um thick P25 brain sections following the ACD protocol for the multiplex fluorescent kit (version 2, Cat. 323100) with the following modifications: (1) sections were treated with 1× Target Retrieval Reagent at ~70 °C for 5 min and (2) were later treated with Protease IV for 30 min at 40 °C. The following RNAscope probe was used: Mm-*Tsc1* (Cat. 591301). After in situ hybridization, immunohistochemistry was performed with the rabbit DsRed antibody (1:500, Cat. 632496, Takara, Mountain View, CA, USA). Lastly, slides were mounted with Vectashield mounting media containing DAPI (Cat. H-1800).

### 2.4. Cell and Synapse Counting

For assessing VIP and tdTomato cell densities, 10× images were taken at the somatosensory cortex at postnatal ages from two or three nonadjacent sections and from both hemispheres for each replicate. A box of a defined area was drawn over a region of interest. Cells were counted within that box and were divided by the box area. For lamination counts, we used DAPI to subdivide neocortical layers.

For tallying cC3^+^/tdTomato^+^ cells, we counted from every 5th section, which covers 25 P25 coronal sections across the rostral-caudal extent of the neocortex. From each mouse brain, we counted over 3000 tdTomato+ cells from WT and cHet brains and over 5000 tdTomato+ cells in the cKO.

For synapse counting, confocal image stacks (0.25 um step size) were processed with ImageJ. Background subtraction was applied, and we colocalized the channels. We counted the number of synapses that colocalized with vGlut1 and PSD95 or nAChR α4 and VAChT in each focal plane. Synapse numbers were then normalized to the length of the dendrite or to the circumference of the soma. The length of the dendrite was determined by the segmented line that was drawn over the dendrite where synapses were counted. Circumference and soma area were determined by an enclosed polygon that was drawn around the soma.

### 2.5. Electrophysiology

Acute cortical slice preparation: Adult mice of either sex (P45–P60 days) were anesthetized with an intraperitoneal injection of euthasol and transcardially perfused with an ice-cold cutting solution containing (in mM) 210 sucrose, 2.5 KCl, 1.25 NaH_2_PO_4_, 25 NaHCO_3_, 0.5 CaCl_2_, 7 MgCl_2_ and 7 dextrose (bubbled with 95%_O2_–5% CO_2_, pH ~7.4). Mice were decapitated, the brains were removed, and two parallel cuts were made along the coronal plane at the rostral and caudal ends of the brains. Brains were mounted on the flat surface created at the caudal end. Approximately three coronal slices (250 μm thick) were obtained using a vibrating blade microtome (VT1200S, Leica Microsystems Inc.). Slices were allowed to recover at 34 °C for 30 min followed by 30 min recovery at room temperature in a holding solution containing (in mM) 125 NaCl, 2.5 KCl, 1.25 NaH_2_PO_4_, 25 NaHCO_3_, 2 CaCl_2_, 2 MgCl_2_, 12.5 dextrose, 1.3 ascorbic acid and 3 sodium pyruvate.

Whole-cell patch clamp recordings: Somatic whole-cell current clamp and voltage clamp recordings were obtained from submerged slices perfused in heated (32–34 °C) artificial cerebrospinal fluid (aCSF) containing (in mM) 125 NaCl, 3 KCl, 1.25 NaH_2_PO_4_, 25 NaHCO_3_, 2 CaCl_2_, 1 MgCl_2_ and 12.5 dextrose (bubbled with 95% O_2_/5% CO_2_, pH ~7.4). Neurons were visualized using DIC optics fitted with a 40x water-immersion objective (BX51WI, Olympus microscope). tdTomato expressing VIP+ CINs located in layers II/III were targeted for patching. Patch electrodes (2–4 MΩ) were pulled from borosilicate capillary glass of external diameter 1 mm (Sutter Instruments, Novato, CA, USA) using a Flaming/Brown micropipette puller (model P-2000, Sutter Instruments). For current-clamp recordings, electrodes were filled with an internal solution containing the following (in mM): 120 K-gluconate, 20 KCl, 10 HEPES, 4 NaCl, 7 K2-phosphocreatine, 0.3 Na-GTP and 4 Mg-ATP(pH ~7.3 adjusted with KOH).

For voltage-clamp recordings, the internal solution contained the following (in mM): 130 Cs-methanesulfonate, 10 CsCl, 10 HEPES, 4 NaCl, 7 phosphocreatine, 0.3 Na-GTP, 4 Mg-ATP and 2 QX314-Br (pH ~7.3 adjusted with CsOH). Electrophysiology data were recorded using a Multiclamp 700B amplifier (Molecular Devices, San Jose, CA, USA). Voltages have not been corrected for measured liquid junction potential (~8 mV). Upon successful transition to the whole-cell configuration, the neuron was given at least 5 min to stabilize before data were collected. Series resistance and pipette capacitance were appropriately compensated before each recording. Series resistance was usually 10–20 MΩ, and experiments were terminated if series resistance exceeded 25 MΩ.

Electrophysiology protocols and data analysis: All data analyses were performed using custom routines written in IGOR Pro (Wavemetrics, Portland, OR, USA). The code is available upon request. Resting membrane potential (RMP) was measured as the membrane voltage measured in current clamp mode immediately after reaching the whole-cell configuration. Input resistance (Rin) was calculated as the slope of the linear fit of the voltage–current plot generated from a family of hyperpolarizing and depolarizing current injections (−50 to +20 pA, steps of 10 pA). Firing output was calculated as the number of action potentials (APs) fired in response to 800 ms long depolarizing current injections (25–500 pA). Firing frequency was calculated as the number of APs fired per second. Rheobase was measured as the minimum current injection that elicited spiking. Firing traces in response to 50 pA current above the rheobase were used for the analysis of single AP properties–AP amplitude and AP half-width. The threshold was defined as the voltage at which the value of the third derivative of voltage with time is maximum. Action potential amplitude was measured from threshold to peak, with the half-width measured at half this distance. The coefficient of variance (CV) for the inter-spike interval (ISI) was calculated as the ratio of standard deviation to the mean. The sag ratio was calculated as the ratio of peak voltage to the steady-state voltage in response to hyperpolarizing current injections (−50 to –250 pA, steps of 50 pA). The rebound slope was measured as the slope of the rebound potential amplitude as a function of the steady-state voltage in response to hyperpolarizing current injections.

Spontaneous excitatory currents were recorded for 5 min with neurons voltage clamped at −70 mV. Currents were analyzed off-line using Clampfit (pClamp, Molecular Devices) event detection.

### 2.6. Immunohistochemistry

For all immunohistochemistry on postnatal brains, staining was carried out on free-floating sections as described previously [27]. For synapse immunohistochemistry, sections were pre-treated with pepsin for 10 min at 37 °C to enhance the staining as described before [28], and sections were incubated with primary antibodies for two days. We performed all immunohistochemistry on *n* ≥ 3 biological replicates for each control and mutant.

### 2.7. In Situ Hybridization

We performed in situ hybridization on a minimum of *n* = 3 biological replicates for each control and mutant. In each case, a rostrocaudal series of at least ten sections was examined. In situ hybridizations were performed using digoxigenin-labeled riboprobes as described previously [29,30]. Vip probe was made using a plasmid with a sequence of Vip cloned in using primers from the Aleen Brain Atlas; 5′ primer: CCTGGCATTCCTGATACTCTTC; 3′ primer: ATTCTCTGATTTCAGCTCTGCC. The plasmid was linearized with HindIII restriction enzyme, and the T7 enzyme was used to make an antisense probe.

### 2.8. Image Acquisition and Analysis

Fluorescent IHC images in Figure 1 were taken using a Coolsnap camera (Photometrics) mounted on a Nikon Eclipse 80i microscope using NIS Elements acquisition software version 4.2.0 (Nikon). Bright-field ISH images in Figure 1 were taken using a DP70 camera (Olympus) mounted on an Olympus SZX7 microscope. Brightness and contrast were adjusted, and images were merged using ImageJ software version ImageJ 2. Synapse IHC images from neocortical layers II/III were taken on an OMX-SR confocal microscope with a 60× objective at 1024 × 1024 pixels of resolution from the Center for Advanced Light Microscopy at UCSF. Synapse IHC images were then aligned and deconvolved before synapse counting.

### 2.9. Statistics

All statistical analyses were carried out on SPSS15 or Graphpad Prism software version GraphPad Prism 9. All data points were found to lie within a normal distribution using a Shapiro–Wilk test and were, therefore, suitable for parametric testing. All data groups were determined to have equal variance as analyzed by Levene’s test. Unless noted in figure legends, all data were assessed by One-Way ANOVA followed by a Tukey’s post-test to determine significance. All data were collected and processed blindly during the data analysis. No data were randomized.

## 3. Results

### 3.1. Conditional Deletion of Tsc1 Increases VIP+ CIN Density in Postnatal Cortex

To test the impact of *Tsc1* loss in VIP+ cortical interneurons (CINs), we crossed *Tsc1*floxed mice [25] with VIP-IRES-Cre mice [26]. The Cre-dependent tdTomato reporter [24] was included in each cross. To validate *Tsc1* deletion, we first performed fluorescent in situ hybridization for *Tsc1* and colocalized with tdTomato+ cells in the somatosensory cortex. We found that *Tsc1* conditional knockouts (cKOs) have reduced *Tsc1* puncta, suggesting effective loss of the *Tsc1* transcript (Figure 1A). We also examined whether the mTOR activation marker, phosphorylated-S6 (pS6), was elevated in cKO cells; there is a significant increase in tdTomato+ cKO CINs co-labeled for pS6 (Figure 1B,C *p* = 0.0005). Consistently, *Tsc1* deletion in VIP+ CINs causes an increase in the soma (cell body) size (Figure 1D; WT vs. cKO *p* < 0.0001; cHet vs. cKO *p* = 0.006).

Next, we assessed VIP+ CIN numbers in the neocortex of WT, cHet and cKO mice. Using VIP immunohistochemistry (IHC), we found a ~2-fold increase in VIP+ CIN density in cKOs at P35 (Figure 2A,B; WT and Het vs. cKO *p* < 0.0001). Because *Tsc1*’s regulation of mTOR signaling can impact protein translation [31], loss of *Tsc1* may affect VIP protein levels. To rule out that the higher VIP+ CIN numbers are due to increased translation of VIP protein and, therefore, an increase in the probability of identifying VIP+ CINs, we assessed VIP RNA transcripts by performing VIP in situ hybridization. As with the VIP IHC, we found an increased density of VIP+ cells in the P35 cKO neocortex (Figure 2A,C; WT vs. cKO *p* = 0.03). Thus, the number of VIP+ CINs is increased when *Tsc1* is lost in the postmitotic VIP¬+ CIN lineage. 

Next, we measured when VIP+ CIN numbers were increased by assessing different developmental ages. VIP-IRES-Cre induces recombination shortly after P0. Thus, we counted the number of tdTomato+ cells from VIP-IRES-CRE Ai14 mice in the neocortex at ages P3, P7, P15, P25, P35 and P50 (example images are shown in Appendix A). In WTs and cHets, we saw tdTomato+ numbers plateau between P7 and P15 and then gradually decrease afterward through P35 (Figure 2D,E; Appendix A). However, in cKOs, we did not see a reduction until after P25. Rather, tdTomato+ numbers remained unchanged from P7 through P35, with this trend still present by P50. We first saw a significant difference in numbers between cKOs and WTs or cHets at age P25 (Figure 2D,E; Appendix A). The difference in WT and cKO numbers became obvious by P25 (Figure 2E; P25 WT vs. cKO *p* = 0.003), and while P15 WT and Het numbers decreased by P25 and P35, the cKOs either did not change or did so at a lower increment (Figure 2E; P15 WT vs. P25 WT *p* = 0.0001 and P35 WT *p* < 0.0001; P15 Het vs. P25 Het *p* = 0.0004 and P35 Het *p* < 0.0001; P15 cKO vs. P35 cKO *p* = 0.02). Furthermore, cannabinoid receptor 1 (CB1) and Calretinin (CR) expression were also elevated in P35 cKOs, markers that are co-expressed in subsets of VIP+ CINs [32,33] (Appendix A; CR cHet vs. cKO *p* = 0.0001; CB1 cHet vs. cKO *p* < 0.01); Sp8+ cells were not grossly changed, perhaps because this marker is expressed in both VIP and other CGE-lineage CINs (Appendix A). Therefore, loss of *Tsc1* in postnatal VIP+ lineages prevented a late developmental reduction of these CINs, which we discovered to occur between P15 and P25.

Because P0 VIP-lineage CINs were still migrating to their final cortical lamina when *Tsc1* was first deleted, it is possible that aberrant migration may account for increased VIP+ CIN numbers in the neocortex. However, we did not see a gross change in the distribution of VIP+ cells in the forebrain at P3, P7 or P25. Moreover, because there were no initial changes in VIP numbers between WTs, cHets and cKOs before P25, we hypothesized that loss of *Tsc1* promoted CIN survival, i.e., decreased apoptosis. Thus, we counted the number of tdTomato+ CINs co-expressing cleaved-caspase 3 (cC3) at P25, the age when there was a significant increase in VIP+ CIN numbers. We saw fewer cC3+ co-labeled CINs at P25 in the cKOs (Figure 2F,F’,G; WT vs. cKO *p* = 0.002; cHet vs. cKO *p* = 0.01). To confirm the cC3 result, we performed a terminal deoxynucleotidyl transferase biotin-dUTP nick end labeling (TUNEL) assay. Likewise, we saw fewer tdTomato+ cells that were TUNEL-positive in P25 cKOs (Appendix A; WT vs. cKO *p* < 0.0001; cHet vs. cKO *p* < 0.0007); higher magnification images of the WT and cHet are shown in Appendix A). Together, these findings show that *Tsc1* is likely necessary for the postnatal apoptosis-mediated reduction of VIP+ CINs between P15 and P25.

### 3.2. Conditional Deletion of Tsc1 Affects the Intrinsic and Synaptic Properties of VIP+ CINs

Using ex vivo slice electrophysiology, we asked whether conditional deletion of a single or both copies of the *Tsc1* gene affects the physiological properties of VIP+ CINs. Because our crosses included Ai14, VIP+ CINs in all genotypes (WT, cHets and cKO) expressed tdTomato. We used this preferential expression of tdTomato to target VIP+ CINs for whole-cell patch clamp recordings. Consistent with previous studies, VIP+ CINs recorded from WT mice had characteristic intermittent firing properties and showed HCN channel-dependent sag and rebound voltages [34,35,36] (Figure 3). While there was no change in resting membrane potentials, the loss of both copies of *Tsc1* significantly reduced the input resistance (Rin) of VIP+ CINs (Figure 3A–C; WT and cHet vs. cKO *p* < 0.0001). Interestingly, HCN channel-mediated rebound voltage was increased in the cKOs, thus suggesting a role of *Tsc1* in the regulation of ion-channel expression and intrinsic membrane properties (Figure 3D–F; WT vs. cKO *p* = 0.05 and cHet vs. cKO *p* = 0.03). We next asked whether *Tsc1* deletion affects the firing output of VIP+ CINs. Previously, we had observed that conditional deletion of *Tsc1* in MGE-derived CINs causes the SST+ CINs to acquire physiological properties characteristic of fast-spiking PV+ CINs [21]. Also, we observed a significant role of *Tsc1* in the regulation of the excitability of SST+ CINs. Here, too, we observed that loss of *Tsc1* increased the maximum firing frequency and spike output of VIP+ CINs (Figure 3G–I; max firing frequency; WT vs. cKO *p* = 0.02; number of spikes; Two-Way ANOVA; genotype x current *p* = 0.0001). We also found a significant increase in action-potential half-width in VIP+ CINs lacking single or both copies of *Tsc1* but no change in amplitude (Figure 3J,K; WT vs. cHet *p* = 0.001 and cKO *p* = 0.004). We compared the variability in inter-spike intervals (ISIs) by calculating the coefficient of variance (CV). Interestingly, in addition to changes in overall firing output, loss of *Tsc1* reduces the CV ISI in VIP+ CINs (Figure 3L; WT vs. cHet *p* = 0.03 and cKO *p* = 0.03). Together, our results suggest a crucial role of *Tsc1* in the regulation of membrane properties, ion-channel expression and excitability of VIP+ CINs.

In addition to changes in membrane properties and ion-channel expression, changes in excitatory synapses can affect the excitability and input-output transformation of cortical neurons. Importantly, we and others have demonstrated that deletion of *Tsc1* affects both excitatory and inhibitory synapses [21,37,38,39]. Therefore, we next assessed how the loss of *Tsc1* affects the excitatory synapses of VIP+ CINs. For this, we obtained voltage-clamp recordings from VIP+ CINs and quantified the frequency and amplitude of spontaneous excitatory postsynaptic currents (sEPSCs) (Figure 4A). The frequency and half-widths of sEPSCs recorded from WTs, cHets and cKOs were not different (Figure 4B,D). However, the amplitude and AUC of sEPSCs recorded from VIP+ CINs in cKOs were significantly higher (Figure 4C,E; amplitude: WT vs. cKO *p* = 0.01; AUC: WT vs. cKO *p* = 0.007). Overall, these findings suggest an important role of *Tsc1* in shaping the excitability of VIP+ CINs via its regulation of the membrane electrical properties and excitatory synapses on VIP+ CINs.

### 3.3. Conditional Deletion of Tsc1 Affects Synaptic Inputs onto VIP+ CINs

Because the loss of *Tsc1* in CGE-derived VIP+ CINs resulted in altered electrical properties, we analyzed whether synapse numbers are affected in VIP+ CINs in cKOs. There are two major synaptic inputs onto VIP+ CINs: glutamatergic and nicotinic acetylcholine synapses. We quantified glutamatergic synapses along the proximal dendrites and in the soma of VIP+ CINs (tdTomato+) by counting puncta expressing vGlut (presynaptic side) and PSD95 (postsynaptic side). Because *Tsc1* deletion in VIP+ CINs caused an increase in the soma (cell body) size (Figure 1D), we normalized synapse numbers in the soma by its circumference. The density of glutamatergic synapses increased in the soma in cKOs compared with WTs and cHets (Figure 5A; WT vs. cKO *p* = 0.01; cHet vs. cKO *p* = 0.005). Moreover, the size of synaptic puncta increased in both cHets and cKOs (Figure 5A; *p* < 0.0001; WT vs. cHet and cKO). However, the density along the proximal dendrite was reduced in both cHets and cKOs (Figure 5B; WT vs. cHet *p* = 0.004; WT vs. cKO *p* = 0.01). We also quantified nicotinic cholinergic synapses in VIP+ CINs by counting puncta expressing vAchT (presynaptic side) and nAchA4R (postsynaptic side). Similarly, the density of nicotinic cholinergic synapses in the soma increased in cKOs compared with cHets (Figure 5C; WT vs. cHet *p* = 0.02; cHet vs. cKO *p* = 0.0005) and the size of synaptic puncta increased in both cHets and cKOs (*p* < 0.0001, WT vs. cHet and cKO). Like before, the density of nicotinic cholinergic synapses along the proximal dendrite was reduced in cHets and cKOs (Figure 5D; WT vs. cHet *p* < 0.0001; WT vs. cKO *p* = 0.0002). These findings demonstrate a critical role of *Tsc1* in the regulation of synaptic input density onto VIP+ CINs. Importantly, the increase in excitatory inputs fits with our observation of higher sEPSC amplitude in cKOs.

## 4. Discussion

This study elucidated molecular, cellular, and electrophysiological changes in the mammalian brain caused by reduced dosage of *Tsc1*, which underlies TSC with comorbid seizures and autism. We uncovered new features in a subgroup of GABAergic inhibitory interneurons, those derived from the CGE and that express VIP. Unique to other interneurons, VIP+ CINs can preferentially synapse onto other inhibitory neurons and lead to disinhibition in the local circuit [20,40], and their dysfunction underlies some local circuit properties as well as a functional target of some syndromic genes [41,42]. Our mutant VIP-lineage CINs displayed an unexpected developmental trajectory when *Tsc1* was deleted, including reduced apoptosis between P15 and P25. Due to the normal disinhibitory properties of VIP interneurons, the increase in mutant CGE CINs likely causes reduced overall circuit inhibition that may contribute to a more excitable brain, which could, in turn, contribute to the high rate of seizures in those diagnosed with TSC.

One of the striking findings was that loss of *Tsc1* led to a greater number of VIP+ CINs in the young adult (P35) neocortex. A previous study showed that VIP+ CINs can die off after P7 [43]. Our data more precisely identified the period of VIP+ CIN apoptosis (between P15 and P25). Because *Tsc1* mutants exhibit more VIP+ cells, this is one manner in which the brain may become more hyperexcitable, as VIP+ CINs provide disinhibition by inhibiting other CINs [19,40]. The VIP+ CIN apoptosis period is later than that of MGE-derived CINs, which peaks at P7 [23]. Excitatory inputs onto VIP+ CINs are known to regulate survival. Thus, it is possible that VIP+ CIN survival is dependent on serotonergic and glutamatergic inputs at P7 [44]. However, it is more likely that the increased VIP+ neurons are due to the pro-survival impacts that increased mTOR activity may provide to these cells.

MTOR interacts with a specific set of proteins to form one of two complexes: MTORC1 and MTORC2. Both complexes can inhibit apoptosis and promote survival in distinct manners. MTORC1 inhibits Bax expression, a death effector [45], while MTORC2 stabilizes the anti-apoptotic protein MCL-1 [46]. Future studies are needed to uncover whether these events are occurring in VIP CINs.

Loss of *Tsc1* function in VIP+ CINs may cause an excitatory/inhibitory imbalance by elevating disinhibition in the local circuit. Increasing excitation by deleting *Tsc1* in VIP+ CINs may be achieved by increasing the number of VIP+ CINs (Figure 2), increasing the number of excitatory synapses on these neurons (Figure 5), increasing the firing rate of VIP+ CINs (Figure 3), and by increasing their sEPSC amplitude (Figure 4). This is especially important, as VIP+ CINs lacking just a single copy of *Tsc1* have some intermediate phenotypes, including elevated AP half-width, variability in interspike intervals and larger synaptic puncta. Thus, defects in VIP CINs may contribute, along with other impacted cells, to some TAND symptoms and seizures in those diagnosed with TSC. Finally, while our studies focused on the brain, we cannot rule out influences from the gut because VIP-Cre expression in gut cells may drive some indirect consequences upon the brain.

Both glutamatergic and cholinergic synapses were increased in the soma while decreased in the dendrite in *Tsc1* mutants, which suggests that *Tsc1* participates in the subcellular localization of synapses. Changes in subcellular synapse localization have not been reported in *Tsc1*/*2* mutant mice. However, MTOR participates in local protein synthesis [47]. Inhibition of MTOR via rapamycin leads to increased Kv1.1 expression on dendrites but not on axons [48] by affecting local protein synthesis. Thus, *Tsc1* may cause a change in subcellular synaptic densities through local MTOR-dependent translation.

Finally, hyperactivity seen in TSC patients and mouse models [49,50] is due to *Tsc1* dysfunction in multiple cell types. In layers II/III of the neocortex, VIP+ CINs are likely to be the CIN most affected because the loss of *Tsc1* in layer II/III Parvalbumin+ and Somatostatin+ CINs did not see a change in GABA receptor-mediated IPSCs [51]. In layer V, *Tsc1* deletion caused reduced inhibitory synaptic output in Somatostatin CINs [21]. In the hippocampus, *Tsc1* haplo-insufficiency in MGE-derived interneurons have impaired synaptic inhibition on pyramidal cells [52]. Moreover, loss of *Tsc1* function led to enhanced AMPA and NMDA receptor synaptic currents in hippocampal neurons [39]. While unique observations, most of these findings would lead to greater excitability in the local circuit, and future studies will explore how the dysfunction of VIP+ CINs may contribute.

## Figures and Tables

**Figure 1 cells-13-00052-f001:**
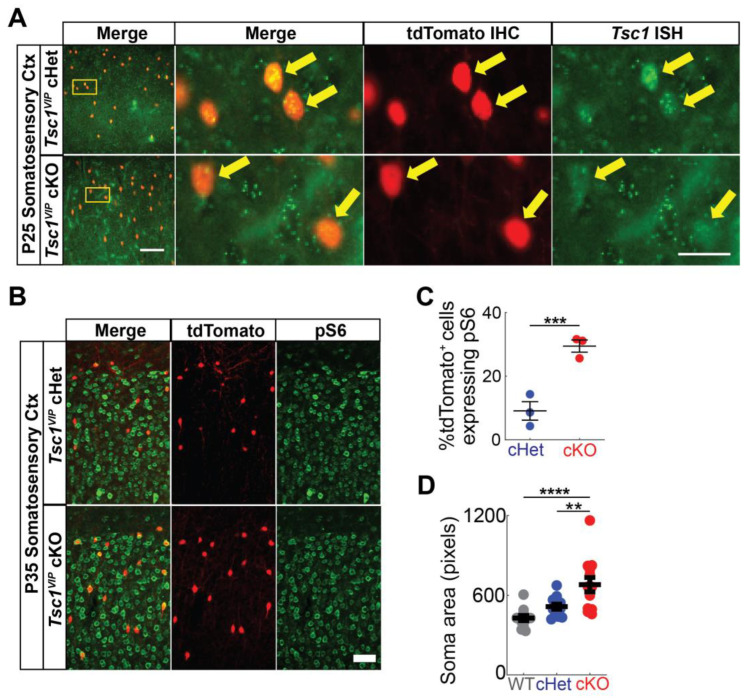
Validation of *Tsc1* deletion and mTOR activation. (**A**) Dual tdTomato immunohistochemistry (red) and *Tsc1* fluorescent in situ hybridization labeling (green) on P25 heterozygote (**top row**) and mutant (**bottom row**) somatosensory cortices. Arrows point to VIP+ (tdTomato+) cells. tdTomato+ cells in *Tsc1* cKOs show reduced *Tsc1* puncta. (**B**) Dual tdTomato (red) and pS6 (green) immunohistochemistry on P35 heterozygote (**top row**) and mutant (**bottom row**) somatosensory cortices. (**C**) Quantitation of percentage of tdTomato+ cells expressing pS6. (**D**) Quantification of cell soma area of VIP+ CINs (tdTomato+) (*n* = 13 somas for each genotype). Data are expressed as the mean +/− SEM; ** *p* < 0.01; *** *p* < 0.001; **** *p* < 0.0001. Scale bars: ((**A**), **left-bottom panel**) 100 µm, ((**A**), **right-bottom panel**) 25 µm and (**B**) 50 µm.

**Figure 2 cells-13-00052-f002:**
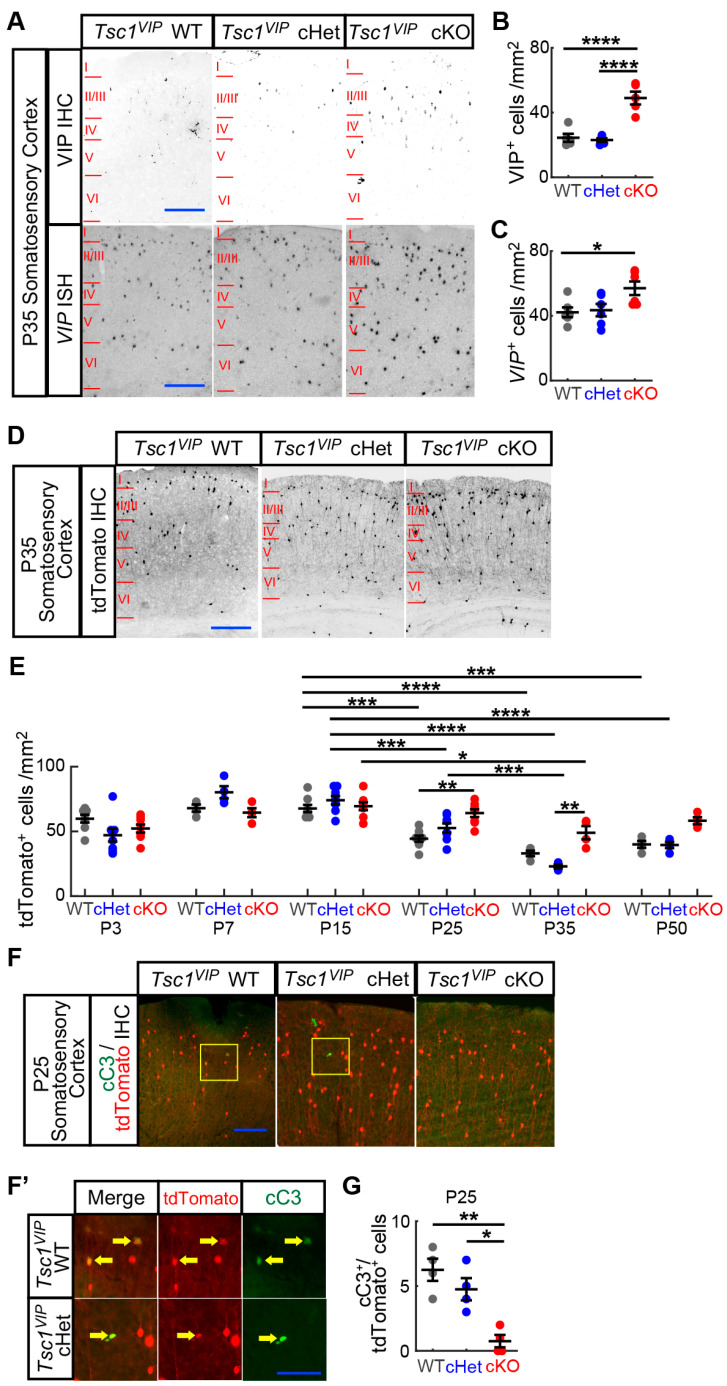
*Tsc1* limits the survival of postnatal VIP CINs. (**A**) VIP immunohistochemistry (**top panels**) (*n* = 5 for each wildtype, heterozygote, and mutant) and VIP in situ hybridization (**bottom panels**) (*n* = 6 for each wildtype, heterozygote and mutant) on P35 wildtype (**left column**), heterozygote (**middle column**), and mutant (**right column**) somatosensory cortices. Scale bar: 50 μm. (**B**,**C**) VIP protein+ (**B**) and VIP RNA+ (**C**) cell density quantifications for all layers in wildtype (grey), heterozygote (blue) and mutant (red) P35 somatosensory cortices. (**D**) tdTomato immunohistochemistry on P35 wildtype (**left**), heterozygote (**middle**) and mutant (**right**) somatosensory cortices. Scale bar: 50 μm. (**E**) tdTomato+ cell density quantification for all layers in wildtype (grey), heterozygote (blue) and mutant (red) P3, P7, P15, P25, P35 and P50 somatosensory cortices (*n* = 8, 4, 8, 8, 4 and 4 for each genotype at P3, P7, P15, P25, P35 and P50, respectively). (**F**) cC3+/tdTomato+ double immunohistochemistry on P25 wildtype (**left**), heterozygote (**middle**) and mutant (**right**) somatosensory cortices. Scale bar: 50 μm. (**F’**) Magnified views from yellow boxes in (**F**). Individual tdTomato (**middle panel**) and cC3 (**right panel**) immunohistochemistry images are shown. Arrow points to VIP+ (tdTomato+) cells that are dying (cC3+). Scale bar: 25 μm. (**G**) cC3+/tdTomato+ double immunohistochemistry cell counts for all layers from rostrocaudal series of coronal P25 WT (grey), cHet (blue) and cKO (red) hemisections; *n* = 4 mice per genotype with over 3000 tdTomato+ cells assessed per brain. Data are represented as mean +/− SEM. * *p* < 0.05; ** *p* < 0.01; *** *p* < 0.001; **** *p* < 0.0001.

**Figure 3 cells-13-00052-f003:**
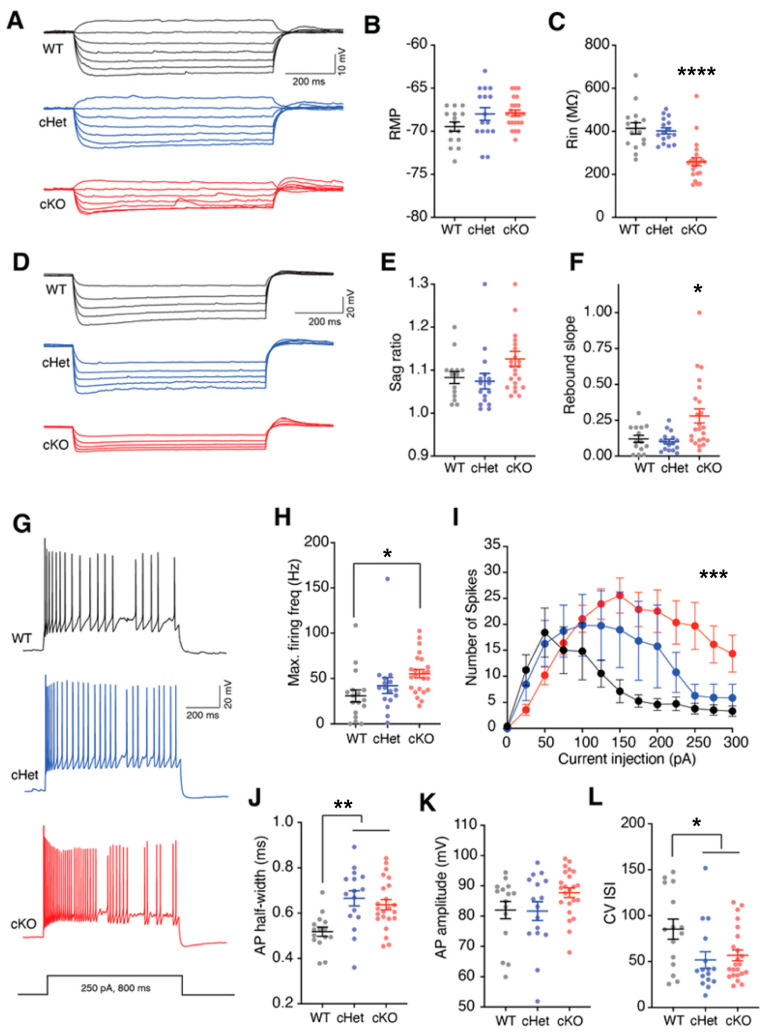
*Tsc1* regulates the intrinsic properties of VIP+ CINs. Example voltage traces (**A**) recorded in response to depolarizing and hyperpolarizing current injections in VIP+ CINs from wildtype (WT, black), condition heterozygous (cHet, blue) or conditional knockout (cKO, red) mice. (**B**) Resting membrane potential of VIP+ CINs from WT, cHet and cKO mice was not different. (**C**) VIP+ CINs in cKOs had significantly lower input resistance (Rin). (**D**) Example voltage traces recorded in response to hyperpolarizing current injections in VIP+ CINs for measuring HCN related properties. (**E**) Sag ratio of VIP+ CINs in WT, cHets and cKOs was not different. (**F**) Significantly larger rebound slopes of VIP+ CINs from cKO mice. (**G**) Example voltage traces showing firing activity in response to depolarizing current injections in VIP+ CINs recorded from WT, cHet and cKO mice. (**H**) Maximum firing frequency was significantly higher in VIP+ CINs recorded from cKOs. (**I**) Number of spikes measured in response to increasing depolarizing currents from VIP+ CINs in WTs, cHets and cKOs is plotted. Note that the number of spikes in cKOs was significantly higher. Two-way ANOVA: Genotype x current interaction, F (24, 612) = 2.937, *p* < 0.0001. (**J**) Action potential (AP) half-width was longer in cHets and cKOs. (**K**) AP amplitude in VIP+ CINs in WTs, cHets and cKOs was not different. (**L**) CV ISI was reduced in VIP+ CINs in cHets and cKOs. * *p* < 0.05; ** *p* < 0.01; *** *p* < 0.001; **** *p* < 0.0001.

**Figure 4 cells-13-00052-f004:**
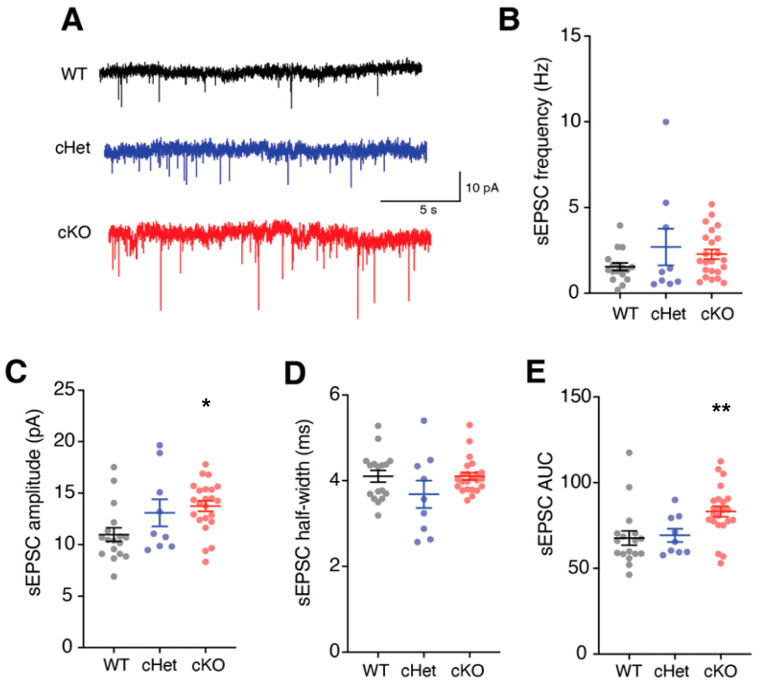
*Tsc1* regulates the excitatory synaptic input currents in VIP+ CINs. (**A**) Example traces showing the spontaneous excitatory postsynaptic currents (sEPSCs) recorded from VIP+ CINs in WT, cHet and cKO mice. (**B**) sEPSC frequency in VIP+ CINs from WTs, cHets and cKOs was not different. (**C**) sEPSC amplitudes were significantly higher in VIP+ CINs from cKO mice. (**D**) sEPSC half-widths in VIP+ CINs from WTs, cHets and cKOs were not different. (**E**) sEPSC area under the curve (AUC) was significantly higher in VIP+ CINs from cKO mice. * *p* < 0.05; ** *p* < 0.01.

**Figure 5 cells-13-00052-f005:**
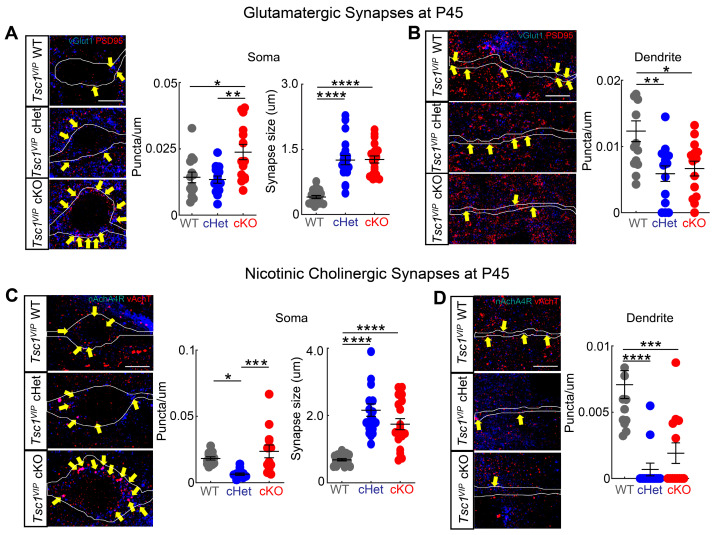
*Tsc1* regulates synaptic density and puncta size in VIP CINs. ((**A**), **left panel**) Single confocal images of wildtype (**top**), heterozygote (**middle**) and mutant (**bottom**), showing the colocalization of vGlut1+ boutons (blue) and PSD95+ clusters (red) onto VIP+ soma (white outline) (yellow arrows) at P45. ((**A**), **right panels**) Quantification of vGlut1+ boutons and PSD95+ clusters colocalization onto the VIP+ soma (*n* = 2 animals, 14 neurons for each genotype) and the size of the boutons (*n* = 2 animals, 20 boutons per group). ((**B**), **left panel**) Single confocal images of wildtype (**top**), heterozygote (**middle**) and mutant (**bottom**), showing the colocalization of vGlut1+ boutons (blue) and PSD95+ clusters (red) onto VIP+ proximal dendrites (white outline) (yellow arrows) at P45. ((**B**), **right panel**) Quantification of vGlut1+ boutons and PSD95+ clusters colocalization onto the VIP+ proximal dendrite (*n* = 2 animals, 14 neurons for each genotype). ((**C**), **left panel**) Single confocal images of wildtype (**top**), heterozygote (**middle**) and mutant (**bottom**) showing the colocalization of vAchT+ boutons (red) and nAchA4R+ clusters (blue) onto VIP+ soma (white outline) (yellow arrows) at P45. ((**C**), **right panels**) Quantification of vAchT+ boutons and nAchA4R+ clusters colocalization onto the VIP+ soma (*n* = 2 animals, 13 neurons for each genotype) and the size of the boutons (*n* = 2 animals, 19 boutons per group). ((**D**), **left panel**) Single confocal images of wildtype (**top**), heterozygote (**middle**) and mutant (**bottom**), showing the colocalization of vAchT+ boutons (red) and nAchA4R+ clusters (blue) onto VIP+ proximal dendrites (white outline) (yellow arrows) at P45. ((**D**), **right panel**) Quantification of vAchT+ boutons and nAchA4R+ clusters colocalization onto the VIP+ proximal dendrite (*n* = 2 animals, 13 neurons for each genotype). * *p* < 0.05; ** *p* < 0.01; *** *p* < 0.001; **** *p* < 0.0001. Scale bars: 5 μm, all panels.

## Data Availability

Data will be provided upon request.

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
