# Peer review of "Tsc1 Loss in VIP-Lineage Cortical Interneurons Results in More VIP+ Interneurons and Enhanced Excitability"

_cells, 2023, doi:10.3390/cells13010052_

Round 1
Reviewer 1 Report
Comments and Suggestions for Authors
In this study (Hu et al. “ Tsc1 loss in VIP-lineage cortical interneurons results in more VIP+ interneurons and enhanced excitability”) the authors investigated changes induced by Tcs1 haplo- or full KO in VIP interneurons. This is a potentially interesting work but I have several questions.
Major comments:
1) Age of animals used for electrophysiological recordings is reported to be between P45 and P60 (L.112). All other results were obtained from P7-P45 mice. What was the reason to use older animals in this case?
2) L150-2. “Input resistance (Rin) was calculated as the slope of the linear fit of the voltage–current plot 151 generated from a family of hyperpolarizing and depolarizing current injections (−50 to 152 +20 pA, steps of 10 pA).”
Although Rin in WT and cHet cells are reported to be not significantly different (Fig. 2C), the current responses (Fig. 2A) to hyperpolarizing pulses look different in these groups (Fig. 2A). Please demonstrate the corresponding I-V curves.
3) L. 154-5 “Firing frequency was calculated as the number of APs fired per second”. This would be acceptable, if the firing patterns were at least similar in the investigated groups. But Fig. 2F does not provide a support to this assumption. WT and cHet cells demonstrate more-or-less regular spiking, while KO cells – definitely not. WT and KO cells show an adaptation, while cHet – not. Please provide a more detailed analysis.
4) Fig. 3A Do synapses in the investigated groups differ in size? This might provide an explanation for the observed difference in sEPSC frequencies. Please show synapse size distributions in three groups.
5) Do sEPSCs in three groups show similar kinetics? Did the authors observe any difference in mEPSC frequency/amplitude/kinetics?
6) For me it is not clear why the authors have included the cHet group into this study. I would expect a comparison in the Discussion. For instance, at P35 the number of dtTomato+ cells is decreased in cHet cortex and increased in KO cortex (Fig 1F)? Maximal firing frequency is increased only in KO cells, but the AP half-width – in both groups (Fig 2I,G)? Action potentials demonstrate adaptation in WT and KO groups ut not in cHet cells (Fig. 2F)? Number of glutamatergic synapses on dendrites is decreased more strongly in cHet group (Fig. 3C), while the number cholinergic soma-located synapses is decreased in cHet group (Fig. 3D)?
These changes have to be at least discussed.
Minor comments:
1) This paragraph (Lines 166-168) is doubled (L. 170-1).
2) Line 209. Scale bar - ???
3) Line 241 - ??
4) Line 252. Do the authors mean P15?
Author Response
In this study (Hu et al. “ Tsc1 loss in VIP-lineage cortical interneurons results in more VIP+ interneurons and enhanced excitability”) the authors investigated changes induced by Tcs1 haplo- or full KO in VIP interneurons. This is a potentially interesting work but I have several questions.
Major comments:
- Age of animals used for electrophysiological recordings is reported to be between P45 and P60 (L.112). All other results were obtained from P7-P45 mice. What was the reason to use older animals in this case?
Response: Our study was aimed at understanding how aberrant development (caused due to Tsc1 loss) shapes the properties of VIP neurons in adult cortical circuits. Since these circuits are still developing between P7-P30, we often wait until 1.5-2 months before measuring electrical activity, which is a standard approach in the field. The earlier ages assessed for cellular and molecular phenotypes were necessary to capture the trajectory of tdTomato+ cells during development.
2) L150-2. “Input resistance (Rin) was calculated as the slope of the linear fit of the voltage–current plot 151 generated from a family of hyperpolarizing and depolarizing current injections (−50 to 152 +20 pA, steps of 10 pA).”
Although Rin in WT and cHet cells are reported to be not significantly different (Fig. 2C), the current responses (Fig. 2A) to hyperpolarizing pulses look different in these groups (Fig. 2A). Please demonstrate the corresponding I-V curves.
Response: Thank you for pointing this out. We have modified the traces in this figure. Panel 2A demonstrates example traces corresponding to I-V curves for input resistance calculation. Panel 2D shows traces in response to hyperpolarizing currents used for sag ratio and rebound slope measurements.
- 154-5 “Firing frequency was calculated as the number of APs fired per second”. This would be acceptable, if the firing patterns were at least similar in the investigated groups. But Fig. 2F does not provide a support to this assumption. WT and cHet cells demonstrate more-or-less regular spiking, while KO cells – definitely not. WT and KO cells show an adaptation, while cHet – not. Please provide a more detailed analysis.
Response: based on the reviewer’s suggestion, we performed an analysis of spike frequency accommodation and coefficient of variance of inter-spike interval. Although SFA values are not significantly different in KOs, we observed significantly reduced CV ISI values in both cHets and cKOs. This suggests that Tsc1 loss affects the variability in firing in an action potential train but does not affect spike frequency accommodation. We have included the CV ISI values in the revised manuscript (Figure 2L).
Figure showing analysis of spike frequency accommodation and inter-spike intervals recorded in VIP+ CINs from WT, cHet, and cKO mice.
- 3A Do synapses in the investigated groups differ in size? This might provide an explanation for the observed difference in sEPSC frequencies. Please show synapse size distributions in three groups.
Response: Glutamatergic and Cholinergic synapse sizes are increased in the soma in cHet and cKO. We have added this result for the three groups to Figures 4B and 4D.
5) Do sEPSCs in three groups show similar kinetics? Did the authors observe any difference in mEPSC frequency/amplitude/kinetics?
Response: We did not observe changes in the width of the sEPSCs but observed significant changes in the area under the curve (AUC) of sEPSPCs. AUC of the postsynaptic currents combines information on the kinetics and the amplitude and gives a general estimation of synaptic strength. These analyses are included in the revised Figure 3E.
We acknowledge that measuring mEPSCs in addition to sEPSCs would be informative. However, this was not part of the initial study design and would require adding substantial new data.
6) For me it is not clear why the authors have included the cHet group into this study. I would expect a comparison in the Discussion. For instance, at P35 the number of dtTomato+ cells is decreased in cHet cortex and increased in KO cortex (Fig 1F)? Maximal firing frequency is increased only in KO cells, but the AP half-width – in both groups (Fig 2I,G)? Action potentials demonstrate adaptation in WT and KO groups ut not in cHet cells (Fig. 2F)? Number of glutamatergic synapses on dendrites is decreased more strongly in cHet group (Fig. 3C), while the number cholinergic soma-located synapses is decreased in cHet group (Fig. 3D)? These changes have to be at least discussed.
Response: The cHet should always be included because this group is most relevant to those diagnosed with TSC. Including it does not distract from the comparison between WT and cKO but can provide some changes that may be relevant for future studies. We have added some text to the discussion to discuss the cHet when relevant.
Minor comments:
- This paragraph (Lines 166-168) is doubled (L. 170-1).
Response: We have deleted the duplicated text.
- Line 209. Scale bar - ???
Response: Scale bar for Figure 1A (top and bottom panels) is 50um. We have corrected it in the figure legend.
- Line 241 - ??
Response: We deleted the text, this was a typo.
- Line 252. Do the authors mean P15?
Response: This was a typo and should be P25.
Reviewer 2 Report
Comments and Suggestions for Authors
The manuscript entitled, “Tsc1 loss in VIP-lineage cortical interneurons results in more VIP+ interneurons and enhanced excitability” by Jia Sheng Hu and colleagues seeks to determine the role of Tsc1 in cortical interneurons. Tsc1 is a member of the heterotrimeric GTPase activating protein formed by the encoded hamartin protein and tuberin and TBC1D7. The proteins operate by opposing Rheb activation of mTOR complex 1 (mTORC1). Loss of Tsc1 activates mTORC1. This occurs in the neurodevelopmental disorder TSC. While numerous studies have elegantly demonstrated the role of excitatory cortical neurons in the formation of cortical tubers and focal cortical dysplasia, less is known about the role of cortical interneurons (CINs). Previous reports demonstrate that loss of Tsc1 from GABAergic neurons leads to increased mTORC1 signaling, impaired migration, and loss of CR and NPY but not PV, SST, or VIP positive neurons (doi.org/10.1093/cercor/bhr300). More recent examination of SST and PV interneurons demonstrated that loss of Tsc1 alters the molecular and morphological characteristics of PV to take on SST electrophysiological properties. Here, the authors attempt to determine the role of a specific role of VIP positive CGE derived CINs. The authors performed time course analyses of CINs and demonstrated VIP neurons decrease in control conditions from P15 to P25. However, the number of homozygous but not heterozygous Tsc1 neurons is somewhat elevated and this appears to be caused by lower amounts of layer I CIN induced apoptosis. The authors additionally demonstrate that these Tsc1 null neurons have increased max firing frequency, number of spikes, and action potential half-width. Finally the authors demonstrate that cell size is increased, there are more glutamatergic synapses, and a change in the nicotinic ionotropic acetylcholine innervation in the soma and dendrites. While the authors have not fully discussed the required literature, the data appears rigorously interpreted. I however have a few minor requests.
1. Is there confirmation that Tsc1 is recombined?
2. Is there evidence that mTORC1 is activated?
3. Is the number of apoptotic cells decreased in non-recombined cells? Is there a circuitry based change occurring in the mice that reduces apoptosis throughout the cortex?
4. Could it be that there is a shift in apoptosis? For example, it looks like the trend is still decreasing from P15 to P35. If you drew this out to P45 would there continue to be a reduction?
5. Line 240 has an error in it.
6. Is the HCN change similar to that discussed by Hsieh and the Bordey lab? Should this be discussed? doi: 10.1126/scitranslmed.abc1492
7. How does this fit with the Knoblich lab paper? Do the mice have tubers and lesions? Can this be discussed some?
8. Do the mice have seizures or overt behavioral issues.
Author Response
The manuscript entitled, “Tsc1 loss in VIP-lineage cortical interneurons results in more VIP+ interneurons and enhanced excitability” by Jia Sheng Hu and colleagues seeks to determine the role of Tsc1 in cortical interneurons. Tsc1 is a member of the heterotrimeric GTPase activating protein formed by the encoded hamartin protein and tuberin and TBC1D7. The proteins operate by opposing Rheb activation of mTOR complex 1 (mTORC1). Loss of Tsc1 activates mTORC1. This occurs in the neurodevelopmental disorder TSC. While numerous studies have elegantly demonstrated the role of excitatory cortical neurons in the formation of cortical tubers and focal cortical dysplasia, less is known about the role of cortical interneurons (CINs). Previous reports demonstrate that loss of Tsc1 from GABAergic neurons leads to increased mTORC1 signaling, impaired migration, and loss of CR and NPY but not PV, SST, or VIP positive neurons (doi.org/10.1093/cercor/bhr300). More recent examination of SST and PV interneurons demonstrated that loss of Tsc1 alters the molecular and morphological characteristics of PV to take on SST electrophysiological properties. Here, the authors attempt to determine the role of a specific role of VIP positive CGE derived CINs. The authors performed time course analyses of CINs and demonstrated VIP neurons decrease in control conditions from P15 to P25. However, the number of homozygous but not heterozygous Tsc1 neurons is somewhat elevated and this appears to be caused by lower amounts of layer I CIN induced apoptosis. The authors additionally demonstrate that these Tsc1 null neurons have increased max firing frequency, number of spikes, and action potential half-width. Finally the authors demonstrate that cell size is increased, there are more glutamatergic synapses, and a change in the nicotinic ionotropic acetylcholine innervation in the soma and dendrites. While the authors have not fully discussed the required literature, the data appears rigorously interpreted. I however have a few minor requests.
- Is there confirmation that Tsc1 is recombined?
Response: We have performed new experiments to assess Tsc1 transcript in tdTomato+ cells and found that cKO tdTomato+ cells exhibit a loss of transcript compared to controls. Part of new Figure S1.
- Is there evidence that mTORC1 is activated?
Response: We stained for the mTOR activity marker, phosphorylated S6 in the somatosensory cortex and found that VIP-Cre+; Tsc1 cKO had increased numbers of pS6 positive labeling. This is part of new Figure S1.
- Is the number of apoptotic cells decreased in non-recombined cells? Is there a circuitry based change occurring in the mice that reduces apoptosis throughout the cortex?
Response: We counted the number of non-VIP+ cells that were cC3+ and found that cKOs had reduced apoptotic cells compared to wildtypes and cHets (widtype: 60.1 +/- 5.2 cC3+ cells, heterozygote: 54.9 +/- 8.1 cC3+ cells, 29.0 +/- 4.5 cC3+ cells; data not shown). It is possible that there is a circuitry-based change in cKO mice that leads to reduced apoptosis that will be investigated in future experiments.
- Could it be that there is a shift in apoptosis? For example, it looks like the trend is still decreasing from P15 to P35. If you drew this out to P45 would there continue to be a reduction?
Response: We examined P50 brains as well and the levels in each genotype compared to P35 appear to have leveled out. See new data in Figure 1E and Figure S2.
- Line 240 has an error in it.
Response: This was a typo and has been deleted.
- Is the HCN change similar to that discussed by Hsieh and the Bordey lab? Should this be discussed? doi: 10.1126/scitranslmed.abc1492
Response: We are not sure as we don't even know which HCN channel may be impacted in our mutants.
- How does this fit with the Knoblich lab paper? Do the mice have tubers and lesions? Can this be discussed some?
Response: These mice do not have any tubers or lesions. We chose to not bring up the Knoblich paper because our manipulation deletes Tsc1 around P0, when these CGE-lineage cells have already become postmitotic and mostly migrated to their final destinations, precluding any progenitor/early developmental perturbations. Thus, our study can’t make any comparisons to other studies where Tsc1 would have been altered in progenitors.
- Do the mice have seizures or overt behavioral issues.
Response: We have not observed overt seizures or behavioral in the mutants.
Reviewer 3 Report
Comments and Suggestions for Authors
The manuscript by Jia Shen Hu and collaborators entitled “Tsc1 loss in VIP-lineage cortical interneurons results in more VIP+ neurons and enhanced excitability” reports results on the status of cortical VIT+ interneurons in the context of a conditional mutation of the TOR-regulator Tsc1. The authors show electrophysiological data consistent with a hyper-excitability of cortical networks. The data and observations are potentially interesting and technically sound, with the exception of the apoptotic data. Overall, however, the interpretation of the results and the discussion are poor, and I do not recommend publication in the present form.
MAJOR ISSUES
The Deletion of the Tsc1 gene should result in a hyper-activity of the TOR pathway. Clearly, both Tsc1 and all many components of the TOR pathway are quite ubiquitous and not restricted to the VIP neurons. When claiming that this conditional mutation prevents an apoptotic pathway, the authors make a statement that is – ON ONE SIDE EXPECTED given the known roles of TOR path – but on the other side NOT REALLY UNDERSTOOD considering the VIP neurons. The only image that is provided is a very poor one, in Figure 1, of double fluorescence of Caspase-3 IFL plus the dtTomato. In this figure I only see 1 single double-positive cell in the heterozygous sample, none in the WT and none in the homozygous. I understand that these events are few, given the rapidity of the apoptotic process. But then, how could this be quantified, given such a low numerosity ?
Can the authors provide another method to show the reduced apoptosis of Tsc1 null VIP neurons ?
Does a somatic/germline Tsc1 mutant (not this conditional in VIP neurons) also display reduced apoptosis, that can be efficiently quantified (e.g. with Annexin FACS) ?
In addition, the finding of reduced apoptosis in the presence of a loss of Tsc1 should be discussed ? Considering the vast literature on the many functions and multiple regulations of the two main TOR complexes.
The authors push the idea that VIP neurons, being affected by the loss of Tsc1, are hyperactive, and that this results in a de-inhibition of the network, hence a propensity to epileptic seizure. Although logical and reasonable, it is surprising that the authors did not TEST this possibility. Are the Tsc1-conditional mutant mice are more susceptible to epileptic seizure upon treatment – for example – with proepileptic drugs, or stress, compared to control animals ? This would be a significant piece of work and significantly add to the quality of the manuscript.
The images in Figure 3 are of poor quality, difficult to interpret. The authors should discuss the significance of having an increased number of soma-associated punctae, and a concomitant reduced number on dendrite-associated punctae. What is the expected consequence of this selective defect ? Are there other examples in literature ? And how is this possibly happening ? How the network UPSTREAM of the VIP neurons could be affected ?
It is not clear what Figure Suppl 2 is useful for. This is a partial quantification of CR and CB1 expressing cells, in an area (the prefrontal cortex) that is not the neocortex, subject of all the rest of the paper. I also note that the localization of the CB1 signal is histologically different between WT and mutant. Is this a phenotype ? or an artifact ?
Furthermore, as stated in the INTRO, the VIP neurons should derive from progenitors residing the Caudal Eminence. Are we sure that these CB1 positive and CR positive cells in the PFC are the same CGE-derived VIP neurons, subject of this study ?
Another aspect that I find not fully convincing : the electrophysiological analyses the authors carried out measure action potential, firing frequency and pattern, responses to current injection, and sEPSC. All these are rather intrinsic properties of the VIP neuron examined, and/or UPSTREAM connection properties. It is reasonable to expect that these may change, due to the mutation. However in order to attribute a function of Tsc1 in the control of the network excitability what is MORE RELEVANT should be the OUTPUT of these neurons onto other neurons. Has any output connection properties or evoked potential being measured ?
I find not really correct that results from a conditional deletion - in a very specific and selected cell population - are being used to derive conclusions on the activity of network. In patients, the TSC1 mutation/LOF certainly is NOT cell type specific, and TSC1 is widely expressed, not only in the brain. So, TSC1 mutations certainly impact on the networks at multiple levels, affecting several neuronal population – both excitatory and inhibitory - and possibly non-neuronal cells. The increased number of VIP neurons is but a limited aspect of the whole story, not a major one. The authors should be cautious about their statement on network function, and consider their data comprehensively with other data on the same mouse model. What is the status of the other CIN types ? What is the activity of the network following loss of Tsc1 somatic/germline ? These would help structure the discussion in a more convincing way.
Minor ISSUES.
At one point in the text and figure the authors use the term “Ai14” to actually indicate “dtTomato” fluorescence, which is confusing. Ai14 is the name of the mouse strain. Please use “dtTomato” to indicate the red fluorescence due to the reporter allele present in the Ai14 mouse strain, both in figures and text.
In the discussion, the authors claim that the heterozygous Tsc1 mutant animals have an intermediate phenotype when compared to the homozygous ones. This is not totally true: in some case yes, in some case no. Please reconsider this statement.
VIP is also a gastro-intestinal active peptide. Is the Tsc1 deletion also occurring in these gut-associated VIP cells ? and is it leading to reduced apoptosis and the onset of phenotypes ?
Author Response
The manuscript by Jia Shen Hu and collaborators entitled “Tsc1 loss in VIP-lineage cortical interneurons results in more VIP+ neurons and enhanced excitability” reports results on the status of cortical VIT+ interneurons in the context of a conditional mutation of the TOR-regulator Tsc1. The authors show electrophysiological data consistent with a hyper-excitability of cortical networks. The data and observations are potentially interesting and technically sound, with the exception of the apoptotic data. Overall, however, the interpretation of the results and the discussion are poor, and I do not recommend publication in the present form.
MAJOR ISSUES
- The Deletion of the Tsc1 gene should result in a hyper-activity of the TOR pathway. Clearly, both Tsc1 and all many components of the TOR pathway are quite ubiquitous and not restricted to the VIP neurons. When claiming that this conditional mutation prevents an apoptotic pathway, the authors make a statement that is – ON ONE SIDE EXPECTED given the known roles of TOR path – but on the other side NOT REALLY UNDERSTOOD considering the VIP neurons. The only image that is provided is a very poor one, in Figure 1, of double fluorescence of Caspase-3 IFL plus the dtTomato. In this figure I only see 1 single double-positive cell in the heterozygous sample, none in the WT and none in the homozygous. I understand that these events are few, given the rapidity of the apoptotic process. But then, how could this be quantified, given such a low numerosity ?
Response: We have new data, Figure S1, showing increased numbers of pS6+ VIP-Cre+ cells in the cKOs, indicating that the mTOR pathway is affected in VIP+ neurons. We overcome the low numbers by counting the number of cC3+; tdTomato+ double positive cells across 25 coronal sections that covers the full rostral-caudal extent. This method is similarly done in Southwell et al., Nature. 2012. In this revision, we clarified how we overcame this low number in the Methods section.
- Can the authors provide another method to show the reduced apoptosis of Tsc1 null VIP neurons? Does a somatic/germline Tsc1 mutant (not this conditional in VIP neurons) also display reduced apoptosis, that can be efficiently quantified (e.g. with Annexin FACS) ?
Response: Other methods of counting cell death such as Annexin V and TUNEL staining will not be a better readout of apoptosis since the whole apoptosis process and the removal of dead cells are quick. Doing FACs with Annexin V is a good idea but will yield high background of cell death because this process involves physical trituration and the disruption of contact between neocortical neurons. These will cause artefactual cell death numbers and dilute the number of cells undergoing Tsc1-depedent apoptosis.
- In addition, the finding of reduced apoptosis in the presence of a loss of Tsc1 should be discussed? Considering the vast literature on the many functions and multiple regulations of the two main TOR complexes.
Response: We have added a new paragraph in the Discussion talking about the reduction in apoptosis in Tsc1 cKO in relation to what is known about mTORC1 and mTORC2.
- The authors push the idea that VIP neurons, being affected by the loss of Tsc1, are hyperactive, and that this results in a de-inhibition of the network, hence a propensity to epileptic seizure. Although logical and reasonable, it is surprising that the authors did not TEST this possibility. Are the Tsc1-conditional mutant mice are more susceptible to epileptic seizure upon treatment – for example – with proepileptic drugs, or stress, compared to control animals? This would be a significant piece of work and significantly add to the quality of the manuscript.
Response: We did not observe any gross seizures. However, studying network activity and susceptibility to epileptic seizures in this Tsc1 cKO is the subject of our next paper.
- The images in Figure 3 are of poor quality, difficult to interpret. The authors should discuss the significance of having an increased number of soma-associated punctae, and a concomitant reduced number on dendrite-associated punctae. What is the expected consequence of this selective defect ? Are there other examples in literature ? And how is this possibly happening ? How the network UPSTREAM of the VIP neurons could be affected ?
Response: We have added a paragraph in the discussion examples where synapse numbers within different subcellular location are affected and how this might be happening.
- It is not clear what Figure Suppl 2 is useful for. This is a partial quantification of CR and CB1 expressing cells, in an area (the prefrontal cortex) that is not the neocortex, subject of all the rest of the paper. I also note that the localization of the CB1 signal is histologically different between WT and mutant. Is this a phenotype ? or an artifact ?
Furthermore, as stated in the INTRO, the VIP neurons should derive from progenitors residing the Caudal Eminence. Are we sure that these CB1 positive and CR positive cells in the PFC are the same CGE-derived VIP neurons, subject of this study ?
Response: We agree this figure was a distraction. We have now replaced this figure with somatosensory cortex images of CB1 in situ hybridization.
- Another aspect that I find not fully convincing : the electrophysiological analyses the authors carried out measure action potential, firing frequency and pattern, responses to current injection, and sEPSC. All these are rather intrinsic properties of the VIP neuron examined, and/or UPSTREAM connection properties. It is reasonable to expect that these may change, due to the mutation. However in order to attribute a function of Tsc1 in the control of the network excitability what is MORE RELEVANT should be the OUTPUT of these neurons onto other neurons. Has any output connection properties or evoked potential being measured ?
Response: We thank the reviewer for pointing this out. We agree that in addition to studying how Tsc1 shapes the physiology and inputs to VIP CINs, it may play an important role in the synaptic output of these neurons. This is an important future direction and out of the scope of the present study.
- I find not really correct that results from a conditional deletion - in a very specific and selected cell population - are being used to derive conclusions on the activity of network. In patients, the TSC1 mutation/LOF certainly is NOT cell type specific, and TSC1 is widely expressed, not only in the brain. So, TSC1 mutations certainly impact on the networks at multiple levels, affecting several neuronal population – both excitatory and inhibitory - and possibly non-neuronal cells. The increased number of VIP neurons is but a limited aspect of the whole story, not a major one. The authors should be cautious about their statement on network function, and consider their data comprehensively with other data on the same mouse model. What is the status of the other CIN types ? What is the activity of the network following loss of Tsc1 somatic/germline ? These would help structure the discussion in a more convincing way.
Response: We were not trying to conclude that alterations of one cell type by itself can lead to widespread network changes. We were trying to explain how increase VIP+ CINs can contribute to network problems. As you have mentioned, hyperexcitability due to loss of Tsc1 is more likely caused by Tsc1 dysfunction in multiple cell types. We acknowledge this notion in the last paragraph of our discussion. Nevertheless, we modified that paragraph by explicitly stating that Tsc1 affecting network activity is due to its loss in multiple cell types.
Minor ISSUES.
- At one point in the text and figure the authors use the term “Ai14” to actually indicate “dtTomato” fluorescence, which is confusing. Ai14 is the name of the mouse strain. Please use “dtTomato” to indicate the red fluorescence due to the reporter allele present in the Ai14 mouse strain, both in figures and text.
Response: Figures have been changed to use tdTomato.
- In the discussion, the authors claim that the heterozygous Tsc1 mutant animals have an intermediate phenotype when compared to the homozygous ones. This is not totally true: in some case yes, in some case no. Please reconsider this statement.
Response: We have changed the text to highlight the exact measures where this occurs and not by being general about it for clarity.
- VIP is also a gastro-intestinal active peptide. Is the Tsc1 deletion also occurring in these gut-associated VIP cells ? and is it leading to reduced apoptosis and the onset of phenotypes ?
Response: We do not know what the effect is concerning VIP+ populations in the gut upon VIP+ CINs in the brain. This is a great question and we added a line in the discussion to bring up this point.
Round 2
Reviewer 1 Report
Comments and Suggestions for Authors
Authors have clarified all my concerns and questions.
I do not have any outher question.
Author Response
Thank you.
Reviewer 3 Report
Comments and Suggestions for Authors
R We have new data, Figure S1, showing increased numbers of pS6+ VIP-Cre+ cells in the cKOs, indicating that the mTOR pathway is affected in VIP+ neurons. We overcome the low numbers by counting the number of cC3+; tdTomato+ double positive cells across 25 coronal sections that covers the full rostral-caudal extent. This method is similarly done in Southwell et al., Nature. 2012. In this revision, we clarified how we overcame this low number in the Methods section.
The addition to the Materials Methods section is appreciated. Also the addition of data on phospho S6+ I VIP+ cells (Figure S1) is appreciated. Since this is quite an important point, the data should be incorporated in the manuscript as a Published Figure, and NOT supplementary.
Figure 1 instead has not really been changed and the authors only show 1 single positive cells, and this is seen only in the Het. Not at all a valid representation. We need to see 1 – 2 apoptotic cells/genotype.
R. Other methods of counting cell death such as Annexin V and TUNEL staining will not be a better readout of apoptosis since the whole apoptosis process and the removal of dead cells are quick. Doing FACs with Annexin V is a good idea but will yield high background of cell death because this process involves physical trituration and the disruption of contact between neocortical neurons. These will cause artefactual cell death numbers and dilute the number of cells undergoing Tsc1-depedent apoptosis.
I am not so sure about this explanation. Instead of physical trituration, that would clearly lead to artifacts, perhaps sequential and more gentle protease digestion might work. Have the authors at least tried ?
Q We did not observe any gross seizures. Studying network activity and susceptibility to epileptic seizures in this Tsc1 cKO is the subject of our next paper
The authors state that studying network activity and susceptibility to epileptic seizures in this Tsc1 cKO is the subject of our next paper. And why so ? These piece of data, when available, would perfectly fit here. Since this paper pushes the idea of a reduced inhibitory action of the VIP-based neuronal network on other interneurons, thus resulting in hyperactivity, an increased susceptibility to pro-epileptic drugs would be a tremendously useful and convicing addition, here.
Q The images in Figure 3 are of poor quality, difficult to interpret. The authors should discuss the significance of having an increased number of soma-associated punctae, and a concomitant reduced number on dendrite-associated punctae. What is the expected consequence of this selective defect ? Are there other examples in literature ? And how is this possibly happening ? How the network UPSTREAM of the VIP neurons could be affected ?
The authors write that “We have added a paragraph in the discussion examples where synapse numbers within different subcellular location are affected and how this might be happening” I cannot find this added sentence ! Or perhaps it is too criptic.
Instead this sentence has been added : “Unique to other interneurons, VIP+ CINs can preferentially synapse onto other inhibitory neurons and lead to disinhibition in the local circuit and their dysfunction underlies some local circuit properties as well as a functional target of some syndromic genes. Our mutant VIP-lineage CINs displayed an unexpected developmental trajectory when Tsc1 was deleted, including reduced apoptosis between P15 and P25”.
First, the sentence is actually two sentences, talks about TWO things, and thus belong to separate paragraphs. Please split and find the appropriate place, for the sake of the flow.
More importantly, the meaning of “…..underlies a functional target of some syndromic genes” in this context it totally foggy and unclear. What is the meaning ? They are talking about local circuits, not syndromic genes.
ABOUT MINOR ISSUES.
Q At one point in the text and figure the authors use the term “Ai14” to actually indicate “dtTomato” fluorescence, which is confusing. Ai14 is the name of the mouse strain. Please use “dtTomato” to indicate the red fluorescence due to the reporter allele present in the Ai14 mouse strain, both in figures and text.
Figures have been changed to use tdTomato instead of Ai14. However there is still one “Ai14” left, in Supplementary Figure 2
One more detail, in the new materials and method section the authors indicate in situ hybridization with a VIP probe. But there are no data shown for this ISH with this probe.
Author Response
Reviewer 3
R We have new data, Figure S1, showing increased numbers of pS6+ VIP-Cre+ cells in the cKOs, indicating that the mTOR pathway is affected in VIP+ neurons. We overcome the low numbers by counting the number of cC3+; tdTomato+ double positive cells across 25 coronal sections that covers the full rostral-caudal extent. This method is similarly done in Southwell et al., Nature. 2012. In this revision, we clarified how we overcame this low number in the Methods section.
The addition to the Materials Methods section is appreciated. Also the addition of data on phospho S6+ I VIP+ cells (Figure S1) is appreciated. Since this is quite an important point, the data should be incorporated in the manuscript as a Published Figure, and NOT supplementary.
RESPONSE: We have moved the phospho S6+ result from supplemental to new main Figure 1. This result is grouped with the Tsc1 in situ hydrization and cell soma size data to highlight the specific loss of Tsc1 and that mTOR signalling is affected.
Figure 1 instead has not really been changed and the authors only show 1 single positive cells, and this is seen only in the Het. Not at all a valid representation. We need to see 1 – 2 apoptotic cells/genotype.
RESPONSE: We have replaced the representative images for this figure (now Figure 2F and F’). The wildtype image shows two dying VIP+ cells, the heterozygote shows one dying VIP+ cells, and the mutant shows none.
- Other methods of counting cell death such as Annexin V and TUNEL staining will not be a better readout of apoptosis since the whole apoptosis process and the removal of dead cells are quick. Doing FACs with Annexin V is a good idea but will yield high background of cell death because this process involves physical trituration and the disruption of contact between neocortical neurons. These will cause artefactual cell death numbers and dilute the number of cells undergoing Tsc1-depedent apoptosis.
I am not so sure about this explanation. Instead of physical trituration, that would clearly lead to artifacts, perhaps sequential and more gentle protease digestion might work. Have the authors at least tried ?
RESPONSE: For FACs analysis, we have tried multiple times using gentler approaches such as papain on postnatal mouse brains, but we still ran into artefactual problems.
Instead, we performed TUNEL analysis on our sections. Like our cC3+ result, wildtype and heterozygote groups had higher number of VIP+ cells that were TUNEL positive than mutant groups. We have added this data in supplemental (see new Supplemental Figure 3).
Q We did not observe any gross seizures. Studying network activity and susceptibility to epileptic seizures in this Tsc1 cKO is the subject of our next paper
The authors state that studying network activity and susceptibility to epileptic seizures in this Tsc1 cKO is the subject of our next paper. And why so ? These piece of data, when available, would perfectly fit here. Since this paper pushes the idea of a reduced inhibitory action of the VIP-based neuronal network on other interneurons, thus resulting in hyperactivity, an increased susceptibility to pro-epileptic drugs would be a tremendously useful and convicing addition, here.
RESPONSE: We agree this would be a great experiment, however, that was not the main goal of this initial manuscript and we predict it will need additional complex experiments to be fully flushed out. For example, VIP interneurons are a small proportion of cells in the cortex and while these data show they are impacted, it is very likely we would need to manipulate these and other cells in a TSC model to see the real impact of these cells. Thus, a more detailed study is needed, which would take more time necessary for this revision.
Q The images in Figure 3 are of poor quality, difficult to interpret. The authors should discuss the significance of having an increased number of soma-associated punctae, and a concomitant reduced number on dendrite-associated punctae. What is the expected consequence of this selective defect ? Are there other examples in literature ? And how is this possibly happening ? How the network UPSTREAM of the VIP neurons could be affected ?
The authors write that “We have added a paragraph in the discussion examples where synapse numbers within different subcellular location are affected and how this might be happening” I cannot find this added sentence ! Or perhaps it is too criptic.
Instead this sentence has been added : “Unique to other interneurons, VIP+ CINs can preferentially synapse onto other inhibitory neurons and lead to disinhibition in the local circuit and their dysfunction underlies some local circuit properties as well as a functional target of some syndromic genes. Our mutant VIP-lineage CINs displayed an unexpected developmental trajectory when Tsc1 was deleted, including reduced apoptosis between P15 and P25”.
First, the sentence is actually two sentences, talks about TWO things, and thus belong to separate paragraphs. Please split and find the appropriate place, for the sake of the flow.
More importantly, the meaning of “…..underlies a functional target of some syndromic genes” in this context it totally foggy and unclear. What is the meaning ? They are talking about local circuits, not syndromic genes.
RESPONSE: We added the subcellular localization section in the second to last paragraph (starting “Both glutamatergic and cholinergic synapases… ") of the Discussion. We provide an example where subcellular kv1.1 localization was affected (specifically in the dendrites) when MTOR signaling was inhibited (Raab-Graham et al., 2006). However, we could not find a specific example that reports increased synapses in soma but a decrease in dendrites. We imagine that this subcellular change can affect action potential initiation in dendrites and overall firing output of VIP+ neurons, but this is hand-waving.
ABOUT MINOR ISSUES.
Q At one point in the text and figure the authors use the term “Ai14” to actually indicate “dtTomato” fluorescence, which is confusing. Ai14 is the name of the mouse strain. Please use “dtTomato” to indicate the red fluorescence due to the reporter allele present in the Ai14 mouse strain, both in figures and text.
Figures have been changed to use tdTomato instead of Ai14. However there is still one “Ai14” left, in Supplementary Figure 2
RESPONSE: Thank you for noticing this. We have changed the label to “tdTomato”.
One more detail, in the new materials and method section the authors indicate in situ hybridization with a VIP probe. But there are no data shown for this ISH with this probe.
RESPONSE: The ISH probe was used in main Figure 2A, bottom panels.